# Biological Insight and Recent Advancement in the Treatment of Neuroblastoma

**DOI:** 10.3390/ijms24108470

**Published:** 2023-05-09

**Authors:** Zoriamin Rivera, Carlos Escutia, Mary Beth Madonna, Kajal H. Gupta

**Affiliations:** 1Division of Pediatric Surgery, Department of Surgery, Rush University Medical Center, Chicago, IL 60612, USA; 2Division of Surgical Oncology, Department of Surgery, Rush University Medical Center, Chicago, IL 60612, USA

**Keywords:** high-risk neuroblastoma, immunotherapy, tumor microenvironment

## Abstract

One of the most frequent solid tumors in children is neuroblastoma, which has a variety of clinical behaviors that are mostly influenced by the biology of the tumor. Unique characteristics of neuroblastoma includes its early age of onset, its propensity for spontaneous tumor regression in newborns, and its high prevalence of metastatic disease at diagnosis in individuals older than 1 year of age. Immunotherapeutic techniques have been added to the previously enlisted chemotherapeutic treatments as therapeutic choices. A groundbreaking new treatment for hematological malignancies is adoptive cell therapy, specifically chimeric antigen receptor (CAR) T cell therapy. However, due to the immunosuppressive nature of the tumor microenvironment (TME) of neuroblastoma tumor, this treatment approach faces difficulties. Numerous tumor-associated genes and antigens, including the MYCN proto-oncogene (MYCN) and disialoganglioside (GD2) surface antigen, have been found by the molecular analysis of neuroblastoma cells. The MYCN gene and GD2 are two of the most useful immunotherapy findings for neuroblastoma. The tumor cells devise numerous methods to evade immune identification or modify the activity of immune cells. In addition to addressing the difficulties and potential advancements of immunotherapies for neuroblastoma, this review attempts to identify important immunological actors and biological pathways involved in the dynamic interaction between the TME and immune system.

## 1. Introduction

Neuroblastoma is a common childhood tumor that mainly affects young children, especially toddlers, and, with a survival rate in high-risk patients of less than 50%, and is responsible for 15% of all childhood cancer deaths [1,2]. Neuroblastoma originates from neural crest tissue and most commonly manifests on the adrenal glands or thoracic, abdominal, or cervical paraspinal ganglia within the first few years of life [3]. The presenting symptoms are dependent on the degree of involvement of the surrounding tissue, with two thirds of patients experiencing metastasis to the regional lymph nodes [4].

Disease progression varies widely between patients, within a range that includes spontaneous recovery without treatment (age < 1 year) to the development of high-risk metastatic tumors with poor prognosis [5]. Disease progression can be characterized by four stages: localized disease without image-defined risk factors (L1); localized disease with image-defined risk factors (L2); metastatic disease (M); and metastatic disease in children <18 mos with disease spread limited to specific sites (MS) [6]. Further, patients are deemed low-risk and high-risk based on multiple factors including age, tumor histology, and other molecular characteristics regarding the tumor microenvironment (TME) [7]. Notably, less than 10% of high-risk neuroblastoma patients have achieved long term cure, mainly due to the way in which high rates of metastasis and minimal residual disease (MRD) in the bone marrow cause late disease relapse [8].

Over the years, significant attempts have been made to identify somatic mutations in human tumors. Cancer treatment that targets tumor-specific mutations holds the promise of accuracy and efficacy while sparing patients from the short- and long-term side effects of chemotherapy and radiotherapy [9]. However, genome-wide searches have been revealing remarkable variations in the occurrence of mutations among tumor types, from melanomas with relatively frequent alterations to pediatric malignancies such as neuroblastoma with rare abnormalities [10]. Chromosomal aberration is frequent in NB, with segmental chromosomal gains or losses and somatic mutations being linked to high-risk disease. Numerical full chromosomal gains are often detected in low-risk tumors. The lack of mutations is a major setback for those exploring tumor-specific immunity and a huge obstacle for those searching for actionable targets from gene alterations [11].

Treatment for high-risk neuroblastoma involves a combination of induction chemotherapy, most commonly doxorubicin, surgical tumor resection, stem cell transplantation, radiotherapy, and, more recently, immunotherapy [7], as depicted in Table 1. The introduction of immunotherapy in neuroblastoma treatment regimens, specifically the anti-disialoganglioside 2 (GD2) antibody dinutuximab, has significantly improved prognosis and patient outcomes [12]. Still, the unique tumor microenvironment poses challenges in the development of novel and effective immunotherapies [13]. The majority of neuroblastoma tumors are immunologically “cold” and devoid of antitumor immune cells and/or they are infiltrated by immune-suppressive cell types, rather than being ‘hot” tumors which are infiltrated by effector immune cells, such as the cytotoxic T cells that can kill tumors cells [14]. As a result, it is difficult to develop therapies that target specific immunologic mechanisms [15]. Dysregulated cellular signaling and metabolism within tumor cells would affect the TME and the response to immunotherapy. However, demonstrating an active adaptive immunity against neuroblastoma has been challenging, particularly in patients at high risk. Given the unusually high tumor size (both primary and metastatic) and its fast multiplication, which might overwhelm a child’s developing immune system, this is not surprising. Insufficient somatic mutations make neuroblastoma weakly immunogenic. A sophisticated immunosuppressive milieu has also been developed by neuroblastoma, which hinders the maturation of functional T cell immunity [16]. This is a complicated and heterogeneous disease, and a variety of factors, including the patient’s age at diagnosis and the disease’s stage, as well as the tumor’s molecular, cellular, and genetic characteristics, determine whether it will spontaneously regress or spread and become resistant to treatment [11].

Despite advances in therapeutic options for patients, there is an urgent need for the further investigation and development of novel immunotherapy treatments that can target the minimally-immunogenic neuroblastoma tumor microenvironment. The aim of this review is to explore advances in the treatment of neuroblastoma, highlight the key immunologic mechanisms that characterize these pediatric tumors, and, finally, draw attention to the role novel immunotherapy treatments can have on improving prognosis for neuroblastoma patients.

## 2. Immunosuppressive Pathways in Neuroblastoma

The pediatric immune system has a wide variety of efficient mechanisms in place to detect and destroy foreign pathogens and malignancies that pose a threat to the host. Many cancers, including neuroblastoma, have developed unique abilities to suppress and/or evade the host immune system. The type of tumor and the extent of its infiltration into the immune cells has been associated with prognosis [12,30]. High-risk (HR) neuroblastoma tumors are characterized as immunologically “cold” and devoid of antitumor immune cells and/or have been infiltrated by immune-suppressive T-regulatory cell types rather than being “hot” and thus infiltrated by effector immune cells such as cytotoxic T cells and natural killer (NK) cells, which can kill tumors cells, and dendritic cells (DC), which are antigen-presenting and activate T cells [31,32,33]. HR neuroblastoma tumors have less tumor-infiltrating lymphocytes [34] than low-risk (LR) and medium-risk (MR) tumors [9]. These findings are supported by studies that show that HR tumors display lower levels of tumor infiltrating lymphocytes (TIL), NK and DC genetic signatures; higher levels of T-regulatory cells; and worse patient outcomes compared with LR tumors [35]. Importantly, LR patients, characterized by higher TIL density, have demonstrated higher disease-free survival (DFS), event-free survival (EFS) and overall survival (OS) [7].

Neuroblastoma tumors employ a variety of methods in which they suppress TILs and NK and DC cells while increasing the levels of regulatory T cells (Tregs). The immunosuppressive tumor microenvironment (TME) comprises myeloid-derived suppressor cells (MDSCs), stromal cells, tumor-associated macrophages [8] and neuroblastoma cancer cells [36]. MDSCs induce several immunoregulatory mechanisms including disialoganglioside (GD2), transforming growth factor (TGF-B), and the transcriptional coactivator TAZ (transcriptional coactivator with PDZ-binding motif) [1,36].

GD2 suppresses DC antigen presentation and the expression of MHC-I and MHC-II molecules, rendering neuroblastoma tumor cells invisible to cytotoxic CD8+ T lymphocytes, and repressing the ability of CD4+ helper T lymphocytes to mount an immune response via cytokine signaling inhibition [12,36]. Neuroblastoma tumors also express reduced levels of interferon (IFNγ), interleukin (IL-6), IL-8, IL-10, IL-12, and IL-21 [12,36,37]. Table 2 summarizes few of the cytokines and there interactions with the ICIs. Under normal conditions, these cytokines function by amplifying the host’s immune responses against pathogens and cancerous cells [38]. Their reduction in neuroblastoma tumors impedes TIL activation and infiltration, contributing to the low immunogenicity of the neuroblastoma TME [12,36,37]. Treatments that target GD2, such as the anti-GD2 antibody dinutuximab, have been developed and have undergone phase trials in neuroblastoma patients with promising results [7]. This discovery has marked an important breakthrough in pediatric solid tumor research and represents promising potential for further investigation in immunotherapy treatments in oncology.

Transforming growth factor beta (TGF-β) is responsible for cell growth, proliferation, differentiation, and apoptosis [37] in various cancers. Although it has been shown that TGF-β plays an inhibitory, anti-tumor role in early cancer stages, recent studies have revealed that TGF-β also displays cancer-promoting properties, including the impairment of NK function, which in turn enhances tumorigenesis and metastasis while inducing immunosuppression and drug resistance [12,37]. This makes TGF-β a promising target in the development of novel immunotherapies. 

Additionally, TAZ has a major impact on the NB TME’s immunosuppressive environment. Increased TAZ expression induces the expression of immune checkpoint programmed death-1 (PD-1) in both T lymphocytes and NK cells. It has been established that the interaction between programmed death-ligand 1 (PD-L1) and PD1 is linked to decreased lymphocyte proliferation, downregulation of NK function, and the survival of cancer cells [36,39]. Importantly, studies have shown that higher expression of PD-1 and PDL-1 is related to poor prognosis [24]. From a therapeutic point of view, it would be beneficial to silence TAZ expression with Food and Drug Administration-approved drugs that target TAZ signaling [36]. Further, inhibition of the PD-1/PD-L1 immunomodulatory checkpoints molecule (ICMS) mechanism has been used to successfully treat some chemotherapy-resistant cancers in adults [36]. Neuroblastoma cells may not be recognized by CTL due to their poor MHC-I expression, but they should be vulnerable to NK cells as a result [40,41]. NK cells can kill neuroblastoma cells, though in some circumstances pre-activation of isolated NK cells is necessary [42]. Nonetheless, neuroblastoma appears to be shielded from NK-mediated death in patients by additional escape mechanisms that adjust the harmony between activating and inhibitory impulses on NK cells. For instance, the ligands (PVR, nectin-2, MICA, MICB, and ULBPs) for the NK cell activating receptors DNAM-1 and NKG2D are not expressed at high levels in neuroblastoma tumors. Thus, NK cell therapies may prove useful in both stand-alone and combination treatments for NB [43].

**Table 2 ijms-24-08470-t002:** The neuroblastoma tumor microenvironment, its immunologic function and interaction with immune checkpoint inhibitors.

Cytokine	Immunologic Function	Interaction with Immune Checkpoint Inhibitor (ICI)
TNFα	A proinflammatory cytokine primarily produced by activated macrophages, T lymphocytes, and NK cells. TNF plays major roles in bone remodeling, infection control, and leukocyte trafficking [37].	TNF has been shown to induce resistance to immunotherapies and acts as a negative biomarker for prognosis. TNFα increases the expression of PD-L1 in tumor cells. Studies have also shown that a TNF-β blockade combined with ICI, such as anti-PD1, has a better therapeutic effect than ICI therapy alone [37].
IFNγ	A proinflammatory cytokine primarily produced by NK cells, activated T lymphocytes, B-cells, and antigen presenting (AP) cells. IFNγ has many immunomodulatory roles including antiviral and antitumor functioning [37].	IFNγ has an antitumor mechanism targeted by ICIs. It increases tumor immunogenicity, suppresses cancer cell proliferation, increases NK cell cytotoxic functioning, and recruit’s tumor-reactive T cells [37]. Clinic studies have reported increased IFNγ levels following anti-PD-1 ICI therapy and improved prognosis. Further, IFNγ has been shown to be a positive biomarker for successful ICI therapy [37].
IL-6	IL-6 has pro and anti-inflammatory properties. Its function is involved in cell survival and growth, immune system regulation, and carcinogenesis. Importantly, it has been shown to promote tumor transmission [44].	IL-6 has been shown to have a negative role in immunotherapy. It has been reported that increased levels of IL-6 induce the production of myeloid-derived suppressor cells (MDSCs), which promote an immunosuppressive TME. Studies have shown that combining anti-IL6 with ICI treatment, such as anti-PD-1 immunotherapy, fosters increased anti-tumor activity and improved prognosis [44].
IL-8	A proinflammatory cytokine produced by macrophages. Its primary roles are to activate neutrophils stimulated by cellular stresses and stimulate endothelial cell proliferation. IL-8 levels have been shown to reflect tumor burden [44].	The interaction between IL-8 and ICI therapy is unclear. However, studies have reported that increased levels of IL-8 are correlated with longer overall survival (OS) in non-small-cell lung cancer (NSCLC) patients treated with nivolumab, and anti-PD-1 ICI [44].
TGF-β	A proinflammatory cytokine produced by leukocytes and stromal cells. It serves many functions, including driving the differentiation of T helper 17 (Th17) cells and regulating cell growth, proliferation, and apoptosis [32,45].	TGF-β inhibits early cancer cells by inducing cell-cycle arrest and apoptosis. However, TGF-β has been shown to have cancer-promoting properties in later stages. Mouse models used to study urothelial cancer have demonstrated that combining anti-PD-1 therapy with the TGF-β antibody reduced the TGF-β pathway and induced tumor infiltration by cytotoxic T cells, resulting in tumor suppression. Similarly, another study has shown that combing ICI therapy with the TGF-β antibody resulted in improved prognosis compared with monotherapy [37].
IL-17	IL-17 has pro- and anti-inflammatory properties. It induces neutrophil-mediated inflammation while also suppressing autoimmune diseases [46,47]. Importantly, studies investigating colorectal cancer have shown that IL-17 exhibits protumoral properties, especially during the early stages [47].	IL-17 signaling has been shown to promote carcinogenesis. Further, studies have shown that inhibiting signaling from this cytokine slows down oncogenesis initiation, suggesting that IL-17 inhibition may be used to halt the early stages of tumor growth. Importantly, studies utilizing mouse models have found that increased IL-17 levels were correlated with high PD-L1 expression. Further, combining IL-17 and a PD-1 blockade induced higher levels of cytotoxic T cells and tumor regression [48].
Il-21	IL-21 has pro- and anti-inflammatory properties. It regulates various immune cells such as NK and cytotoxic T cells while hindering the pro-inflammatory mechanisms of macrophages. IL-21 also exhibits anti-tumor properties [49].	Studies have found that IL-21 hinders tumor development, especially during early stages [50]. Additionally, more recent studies using mouse models have demonstrated that combining an immune checkpoint blockade, such as anti PD-L1, with IL-21 administration increased antitumor activity, characterized by increased CD8+ T cell proliferation and by increased infiltration by, and memory of, effector T cells [50].

As previously mentioned, the combined immunosuppressive mechanisms of neuroblastoma cancer cells create an immunologically “cold” environment devoid of activated T lymphocytes and NK cells, with T-regulatory cells upregulated. Under normal circumstances, NK cells and cytotoxic T cells play a prominent role in preventing tumor progression and metastasis by inducing tumor cell lysis and inflammation [12]. T-regulatory (Tregs) cells function to modulate the immune system by inhibiting other immune effector cells to prevent the over activation of the immune system. The overexpression of Tregs further inhibits the already immunologically inactive TME by suppressing NK and cytotoxic T cell functioning [51,52]. This is especially seen in high-risk neuroblastoma tumors where there are increasingly higher levels of immunosuppressive stromal cancer cells, MDSCs and Tregs [36,52].

## 3. Immunotherapy Targets in Neuroblastoma

Immunotherapy can fall into two different types (Figure 1), one being active immunotherapy and the other being passive. Active immunotherapy relies on directly attacking cancer cells through the stimulation of the immune system [53]. Passive immunotherapy utilizes the acceptance of an organism’s immune system of antibodies, cytokines, and transformed immune cells to act directly on tumor cells [54]. Immunotherapy treatments that can target neuroblastoma cells have been in development over the course of recent decades.

Cancer immune surveillance is a mechanism by which immune cells recognize and eliminate tumor cells. Neuroblastomas produce a highly immunosuppressive environment that has an effect on the body as well as locally in the tumor. Neuroblastomas employ indirect immunoregulatory mechanisms by chartering immune suppressive agents to dampen the activity of immune system. TGF, galectin-1, MIF, soluble GD2 (sGD2), and arginase-2 are some of the soluble mediators that neuroblastoma cells produce, and which have the ability to inhibit lymphocyte activation [55]. Neuroblastoma cancer cells also produce several other immunosuppressive molecules such as, sMICA, sB7-H6, sHLA-E, sHLA-G, IL-10 and HMGB1 [56]. Neuroblastoma tumors are intermixed with myeloid and stromal cell populations with defective activating functions or enhanced suppressive functions, which can prevent TIL from effectuating an anti-tumor response.

One of the areas in which immunotherapies have become focused is that associated with the passive invocation of the immune system to respond to a tumor in ways that would not occur without outside intervention. These therapies target the activation of immune cells such as T-cells to assist in the aim to provide an anti-tumor effect through the infiltration of the immune cells in the tumor. High-risk neuroblastoma with higher T cell infiltration has been associated with improved survival [57]. Increasing the levels of T cell infiltration into the tumor cells must be a goal for the improvement of patient outcomes.

A form of passive treatment for immunotherapy that has shown promising results is the use of monoclonal antibodies (mAbs). One of these immune therapies focuses on the use of mAbs in order to recognize GD2, which is overexpressed relative to a control in neuroblastoma. GD2 is a disialoganglioside that is ubiquitously expressed on the surface of all neuroblastoma cells (source) [29]. This makes GD2 an appealing target due to its specificity for the disease. Adoption of mAbs such as anti-GD2 in both upfront and relapse treatment protocols has dramatically increased survival rates and altered the landscape for children with high-risk neuroblastoma [28]. Even in patients with relapsed disease, dinutuximab may have a treatment value [58]. The success of dinutuximab has sparked investigations into combination therapy with cytotoxic compounds, as well as cellular immunotherapy with (haploidentical) donor NK cells [59,60,61].

Recent technological advancements have made it feasible to analyze the immune response to patient-specific neoantigens that result from tumor-specific mutations. New evidence suggests that clinical immunotherapies may be more effective when these neoantigens are recognized. However, children probably have few, if any, actionable, mutation-generated, immunogenic tumor neoantigens because most juvenile cancers have a very low tumor mutation burden [62,63]. Cancer-testis antigens, as well as other embryonic or differentiation antigens expressed during development and on children’s cancers but not on normal postnatal tissues, may fall under this category [64]. Studies have shown a correlation between the use of these neoantigens and an increase in T cell activity [65]. Recently researchers have identified PHOX2B, a peptide displayed on the surface of neuroblastoma cells, by their MHC molecules [66]. The oncofetal proteins, expressed during embryonic development, though silenced after birth, may be ideal targets for T cell-based immunotherapies in NB due to their restricted expression outside of the tumor. While the comparatively low tumor mutational burden of neuroblastoma may limit neoantigen expression, developmental antigens may be the target of T cell-based immunotherapy. Given the high sensitivity and efficacy of some of these receptors, as has previously been seen with certain modified TCRs, researchers will need to be cautious of the possibility for antigen cross-reactivity. By administering immunomodulating agents that improve the tumor’s immunogenicity, stimulate antigen presentation, increase the patient’s endogenous tumor-reactive T cells, and suppress the tumor’s immunosuppressive microenvironment, it may also be possible to trigger endogenous tumor-reactive T cells in a patient [67,68,69,70]. Tumor-specific neoantigens lead to personalized immunotherapies for patients. Future development of technology involved in the recognition of these neoantigens can help provide an efficient and rapid response.

Cytokines released by cancer cells or cells in the tumor microenvironment support angiogenesis, tumor cell migration and metastasis, and the development of an immunosuppressive microenvironment. These tumor-promoting effects of cytokines also apply to NB, as depicted in Table 3. IL-6 and VEGF have been further characterized as cytokines that stimulate tumor growth and metastasis, while others, such as IFN-γ, can exert anti-NB activity by inducing tumor cell apoptosis and by inhibiting angiogenesis (Figure 2 and Table 3).

## 4. Role of MYCN in Neuroblastoma

Numerous cancers exhibit deregulation of the MYC family of oncogenes, which includes c-MYC, MYCN, and MYCL, and this is frequently correlated with a poor outcome [87]. MYCN amplification is a significant driver in neuroblastoma and is indicative of a poor prognosis in early-stage tumors [88]. Numerous studies have investigated the relationship between MYCN amplification and TIL infiltration and composition. Given the tight relationship between MYCN amplification status and tumor stage, it is unclear which factor is responsible for the correlation between MYCN amplification and TIL infiltration [89]. Inverse correlation between MYCN amplification and leukocyte infiltration has been reported, demonstrating lesser infiltration of MYCN-A tumors by CD8+ and CD4+ T cells, NK cells, NKT cells, B cells, macrophages, and monocytes [90,91]. The presence of iNKT cells has also been predicted by MYCN overexpression at the RNA level in combination with low CCL2 expression [74]. It is significant to note that, even within the group of HR-neuroblastomas, T cell and cytotoxic cell signatures, as well as CD8+ infiltration, were lower in MYCN-A than in MYCN-NA tumors, indicating that low TIL infiltration is at least partially linked to MYCN amplification [92]. A possible cause of decreased TILs infiltration in MYCN-A could be reduced lymphocyte attraction and activation or reduced immunogenicity. It has been previously shown that there is greater TIL infiltration and cytotoxicity in tumors with a higher mutational load, with tumor mutational load being a significant determinant of tumor immunogenicity [74]. Neuroblastomata are typically known to have a low mutational burden and therefore low immunogenicity [63,93]. As an outcome, it is possible to infer that MYCN-A neuroblastoma may be less susceptible to T cell immune monitoring on a variety of levels.

In recent years, novel approaches in the treatment of neuroblastoma have undergone clinical trials. Recently, multi step treatments have been recommended. These involve treatment with granulocyte-macrophage colony stimulating factor gene (GM-CSF), which enhances the immune activation and enhancement of antibody cell-mediated cytotoxicity (ADCC). Several other clinical trials are in progress in which combinational therapy, including drugs and newly defined immunotherapies, are in progress. Table 4 depicts some of these clinical trials, including those that evaluate CAR T cells that target CD171 [94] in patients with NB, clinical trial using novel combinations like 131-I-MIBG and ch14.18/CHO, targeting GD2, GM-CSF, and GD2-CAR NKT cells are also discussed.

## 5. Are Tumors in Children Any Different? Knowing How the Immune System Differs in Children

Immune response to a tumor is associated with a positive response to immunotherapies, a type of medical treatment that aids your body in fighting cancer. Numerous cancer types now routinely receive immunotherapy as a treatment, but little is known regarding the question of how juvenile patients’ immune systems react to malignancies and how they compare to those of adults. Although many treatments have been developed that successfully combat solid cancers in adult patients, many of those treatments cannot be applied in a pediatric setting. This is due to important differences between adult and pediatric immune systems that occur as the body grows and develops. Further, neuroblastoma is a cancer that is rarely diagnosed in adults; almost 90% of cases are diagnosed in children younger than age 5 [1]. For this reason, it is difficult to apply successful cancer treatments in adults to a cancer that predominantly affects young children. The explanation behind this overrepresentation in younger age groups is due to the origin of the cancerous cells. Neuroblastoma originates from abnormally proliferating neural crest cells in-utero due to spontaneous genetic mutations [52]. These cells mature and become part of the newborn’s sympathetic nervous system, with the tumor developing most commonly in the adrenal glands within the first few years of life [1,52], thus explaining why adults do not develop this specific cancer.

The TME that then thrives in young children can be explained by the relative immaturity of the young immune system. At birth, nearly all T cells in the body are naive T cells that have yet to encounter foreign antigens; there is also a larger quantity of Tregs during this earlier time of life [95]. As the body grows, the immune system matures and gradually develops memory during a life-course of exposure to multiple foreign antigens [52,96]. Additionally, pediatric cancers are known to have a lower mutational burden [97]. Combined with the abundance of Tregs relative to cytotoxic T cells, young children foster a susceptible environment for neuroblastoma tumor cells to thrive within. Importantly, the immunological differences between pediatric and adult patients explain why pediatric tumors are less responsive than adult tumors to immunotherapeutic tumor treatments, particularly treatments that utilize immune checkpoint inhibitors (ICI) such as anti-PD-1/PD-L1 [97]. For example, a study in melanoma patients found that younger patients exhibited higher levels of resistance to anti-PD1 inhibition therapy when compared with adults [98]. The same study also showed younger melanoma patients exhibit higher amounts of Treg genetic signatures and a decreased level of cytotoxic CD8+ T cell populations compared with older patients. These results indicate that modulating the TME in young patients may be key in sensitizing pediatric tumors to PD-1 inhibition and other immune checkpoint inhibitors [98].

Patients are further treated with adoptive chimeric antigen receptor (CAR) T cells, cells that are collected from a donor and engineered to express CAR, which targets a tumor-specific antigen [7,99]. Immune-modulatory drugs, such as lenalidomide, have also been developed; these have a direct or indirect impact on host immune cells [7].

The utilization of cytokines and monoclonal antibody therapies to treat high risk neuroblastomas, in conjunction with consolidation therapy and bone-marrow transplant (BMT), is a novel approach that has improved the long-term survival of neuroblastoma patients [51,100]. The disialoganglioside GD2 is an antigen expressed in different cell types of the nervous system (including neurons, peripheral pain fibers, and melanocytes), and, importantly, is expressed at higher levels in neuroblastoma cells, making it an ideal target for antibody development [51,101].

Anti-GD2 antibodies, such as dinutuximab, have been developed and have undergone phase trials in neuroblastoma patients with promising results [7,51]. Tumor cells are labeled with anti-GD2 antibodies, allowing for their recognition and destruction by immune effector cells. The addition of cytokines to this regimen, such as GM-CSF, has been shown to increase the efficiency of neutrophil-mediated ADCC by lymphocytes in neuroblastoma patients [7,102]. This discovery marked an important breakthrough in pediatric solid tumor research—immunotherapy has given way to a promising era in oncology that will significantly improve the survival and long-term sequelae of neuroblastoma patients.

## 6. Challenges and Future Directions for NB Treatment

As previously discussed, the primary challenge in treating high-risk neuroblastoma is the low immunogenicity of neuroblastoma tumors. It has become evident that both the natural immune response and the effectiveness of immunotherapeutic treatments are hindered by the various immunosuppressive characteristics of the neuroblastoma TME. However, the success of dinutuximab is an important example of the promise that immunotherapy has in treating this cancer, as well as other complex childhood cancers [12,103]. Immunotherapies, specifically immune checkpoint inhibitors, may be key in inducing immunologically “hot” TMEs, thereby eliciting T cell reactivity and memory, allowing for more effective and promising anti-cancer treatments. Importantly, utilizing a combination of personalized immunotherapies that target different subjects simultaneously may be required for improved outcomes and prognosis.

It has been established that TILs serve an important prognostic role in neuroblastoma [104]. Increased infiltration by T lymphocytes is correlated with the recruitment of DCs and NK cells, which correlates with favorable outcomes for neuroblastoma patients due to their interactions with each other and with other immune cells in the TME [57,105]. Additionally, a key relationship has been identified in which the expression of DC and NK cell signatures (THBD and NCR1) in neuroblastoma samples is significantly correlated with the expression of genes that the encode important immunomodulatory checkpoint molecules (ICMS) PD1 (programmed-death receptor expressed on lymphocytes) and PDL1 (programmed-death ligand expressed on cancer cells) [104]. It has been established that the interaction between PDL1 and PD1 is linked to decreased lymphocyte proliferation, and the survival of cancer cells [104]. Inhibition of this specific ICM mechanism has been used to successfully treat some chemotherapy-resistant cancers in adults [101]. Although research on the inhibition of checkpoint molecules as drug therapy for pediatric solid tumors is limited, recent studies have begun to shed light on the relationship between expression of ICMS and prognosis in neuroblastoma. Mouse models have yielded significant trends relating PDL1 and neuroblastoma. Importantly, expression of PDL1 is correlated with poor prognosis, indicating that inhibition of PDL1 leads to tumor cell death. Further, combining inhibition therapy with other chemotherapeutic agents corresponds to a prolonged response [1]. Although research is limited, these trends lay the foundation for the possibility of ICMS to provide viable treatment regimens in pediatric neuroblastoma. Further investigation is needed to establish a mechanistic link between the DC-NK axis, T lymphocyte infiltration, expression of key genetic signatures, the PDL1-PD1 pathway, and prognosis for neuroblastoma patients.

In conclusion, immunotherapy has yielded promising results in the treatment of certain high-risk cancers. In neuroblastoma, however, the low immunogenicity of these tumors combined with the relatively immature immune system makes it challenging to effectively administer treatments with high levels of efficacy. Therefore, future studies must first identify ways in which the immune environment of these pediatric tumors can be characterized and induced to overcome the inhibitory mechanisms at play. Identifying these new therapeutic strategies will therefore allow for the improvement of the long-term efficacy of current treatments and yield novel approaches, thereby improving the long-term prognosis of these young patients.

## Figures and Tables

**Figure 1 ijms-24-08470-f001:**
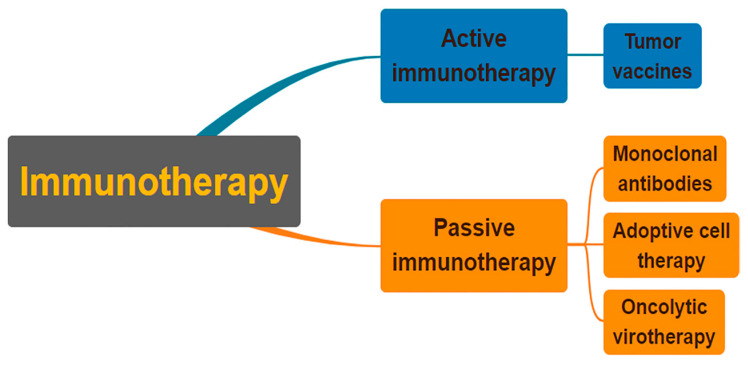
Classification of immunotherapies as either active or passive. In order to combat cancer cells, active immunotherapy stimulates the immune system of the cancer patient. Patients who cannot naturally make immune molecules are given them through passive immunotherapy. Both strategies may utilize a specific or a general strategy.

**Figure 2 ijms-24-08470-f002:**
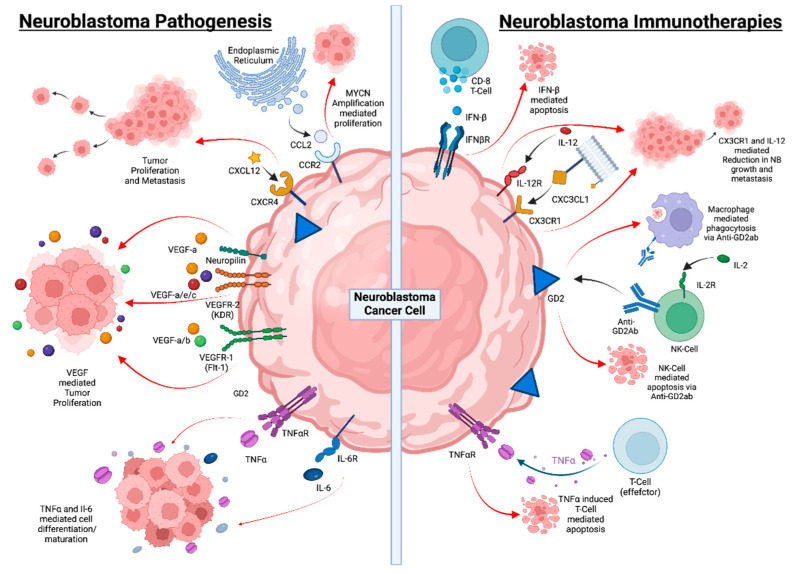
Neuroblastoma progression and underlying mechanisms behind immunotherapy. Neuroblastoma pathogenesis is marked with low MHC-I expression, which makes it difficult for T lymphocytes to recognize neuroblastomas, and low levels of activating ligand expression inhibit immune cell function. T and NK cell activity is further inhibited by soluble immunoregulatory mediators found in TME. However, with a potential immunotherapy, bispecific antibodies may cause T cell tumor reactivity, while anti-GD2 antibodies can activate NK cells, checkpoint inhibition can restore T cell function, and MSC and MDSC can be depleted. Soluble mediators’ immunosuppressive effects can be lessened, which will also boost DC’s ability to co-stimulate. IFN- β has been shown to sensitize NB cells to the cytotoxic effects of chemotherapy drugs such as TMZ. CX3CL1 is another chemokine that is highly expressed on the cell surface of NB tumor cells and combination with IL-12 has been shown to reduce NB growth and eliminate metastasis. Macrophage-mediated phagocytosis and NK-cell-mediated apoptosis using anti-GD2 have been shown to be effective treatments. TNFα induces T-cell-mediated apoptosis of NB cells.

**Table 1 ijms-24-08470-t001:** Recent advances in neuroblastoma therapy.

Treatment	Indication	Clinical Uses and Benefits	Risks and Side Effects	Agents
Surgery	First-line treatment to remove tumor	Results from surgery are often used to determine disease stage if imaging is insufficient and there is a need for additional treatment [17]. For example, if the tumor is in its early stages, it may be possible to completely resect the mass. In other cases, the tumor may not be able to be removed completely, or may have metastasized, requiring the patient to undergo additional treatment such as chemotherapy [18].	Surgical risks largely depend on tumor location, duration of surgery, and the patient’s overall health. Major complications are rare but can include anesthetic errors; bleeding; infection; and damage to organs, nerves, or blood vessels. The risk for complications increases if the tumor is growing into nerves or blood vessels [18].	N/A
Chemotherapy	L1, L2, M, MS First line in cases where the cancer has metastasized too far to be completely removed by surgery	Chemotherapy is often used in conjunction with surgery [19]. Chemotherapy may be used prior to or after surgery. For patients in high-risk groups, a combination of chemotherapy drugs may be used, as well as other therapy regimes. Factors considered are location of the tumor, age, patient health, and others [20].	There are many risks that must be considered and discussed with patients and their families before initiation of chemotherapy [21]. Chemotherapy drugs can be damaging to various cells in the body, leading to side effects. The most common are hair loss, sores, loss of appetite, nausea, vomiting, and GI upset. More severe side effects include leukopenia causing infection; thrombocytopenia leading to excessive bruising and bleeding; and anemia [22].	Cyclophosphamide Cisplatin Vincristine Doxorubicin Etoposide Topotecan Melphalan Busulfan Thiotepa
Radiation Therapy	M, MS Only used in patients considered high-risk	Radiation therapy is necessary for cases where the cancer has spread and is not removable with surgery and chemotherapy alone. It also may be used in patients with emergency, life-threatening symptoms to rapidly reduce the tumor’s size [23]. Two radiotherapy options are available. External beam radiation therapy focuses a radiation beam onto the tumor from a machine source, known as total body irradiation. Iodine meta-iodebenzylfuanidine (MIBG) radiotherapy, which is chemically similar to norepinephrine (NE), may be initially injected into the blood to detect neuroblastoma or to directly deliver radiation to the tumor [24].	Radiation therapy is avoided when possible due to the high risk of various short and long-term side effects. These include skin burns, hair loss and other skin reactions, nausea and GI upset. Long-term effects include bone damage; growth arrest; hypothyroidism; cardiovascular and respiratory problems; and cellular damage, especially to DNA [22]. MIBG therapy typically results in milder side effects due to its local mechanism of action [25].	External beam radiation therapy MIBG radiotherapy
High-Dose Chemotherapy and Stem Cell Transplant (STC)	M, MS used in high-risk patients unlikely to be respond to other treatments	This treatment is a combination of chemotherapy, usually higher doses, and bone marrow stem cell transplantation to replace the bone marrow cells injured by the chemotherapy [21]. Many patients have a second stem cell transplant several months apart. The patient is given a medication called filgrastim (G-CSF) to induce bone marrow cell proliferation, which is then collected and later used for the transplant [26].	High-dose chemotherapy and stem cell transplantation is a complex procedure and comes with several short- and long-term effects. High-dose chemotherapy is severely toxic to the body’s cells and can lead to anemia, bleeding, GI upset, mouth sores, loss of appetite, and hair loss. Long-term effects include damage to various organs including the liver, heart, or lungs; hypothyroidism; poor hormone control; infertility; osteopenia; and higher risk for infections and other cancers such as leukemia [21].	High doses of chemotherapy agents listed above
Retinoid Therapy	M, MS Used after completing high-dose chemotherapy and STC	This treatment uses retinoids. Retinoids are chemical compounds that are structurally related to vitamin A and work as differentiating agents, meaning that they are thought to induce cancer cell differentiation to normal cells [27]. Retinoid therapy is recommended for high-risk neuroblastoma patients that have completed high-dose chemotherapy [28]	The side effects of retinoid therapy are less severe than the other neuroblastoma treatments. The most often reported side effects are cracked lips, joint and muscle pain, and epistaxis [29].	Isotretinoin (13-cis-retinoic acid)

**Table 3 ijms-24-08470-t003:** Cytokines and chemokines in neuroblastoma pathogenesis and preclinical therapy.

Cytokine	Function	Role in Neuroblastoma
VEGF	VEGF acts as a pro-inflammatory cytokine by increasing endothelial cell permeability, by inducing the expression of endothelial cell adhesion molecules, and via its ability to act as a monocyte chemo-attractant [71].	Studies looking at the expression of several markers in NB xenografts have shown that some angiogenic factors including VEGF-A, -B and -C are associated with advanced NB stage [72].
CCL2	Chemokine CCL2 (also known as monocyte chemo-attractant protein-1, MCP-1) is one of the vital chemokines that control the migration and infiltration of monocytes/macrophages [73].	The infiltration of neuroblastoma cells by invariant NKT (iNKT) cells was found to correlate with the expression of the chemokine CCL2 by the tumor [74].
CXCL12	CXCL12 acts through its receptors CXCR4 and CXCR7. CXCR4 stimulation leads to the activation of numerous signaling pathways depending on the associated cell types, while CXCR7 has mainly been shown to be involved in scavenging CXCL12, although it can activate a MAP kinase pathway through β-arrestin in several systems [75].	CXCL12 and CXCR4 have been demonstrated to be overexpressed in NB cell lines in addition to primary metastatic NB. This hints at the role of CXCL12 in its connection to autocrine/paracrine signaling of tumor growth instead of the development of metastatic pathways [76].
CXC3CL1	CXC3CL1 is an unusual chemokine expressed on the cell surface and acting as adhesion molecule by binding to its receptor CX3CR1.7 and is also expressed in a variety of human tissues and cell lines, where it mediates leukocyte migration and adhesion [77].	It has been shown in animal models that CX3CL1 is able to inhibit NB growth and eradicate metastasis when used in combination with IL-12 through the attraction of immune cells to the tumor site [78].
IL-6	IL-6 and VEGF are the best characterized cytokines to stimulate tumor growth and metastasis, while others, such as IFN-γ, can exert anti-NB activity by inducing tumor cell apoptosis and inhibiting angiogenesis [79].	IL-6 is introduced into the bone marrow by the bone marrow stromal cells (BMSC) which promotes the growth and survival of neuroblastoma cells [80].
IL-7	IL-7 is a cytokine that stimulates proliferation of all cells in the lymphoid lineage (B, T and NK cells) [81].	A study using a humanized mouse model of metastatic NB showed that the combinatory therapy of human γδ T cells, hu14.18 anti-GD2 antibody, and Fc-IL-7 was able to increase the survival rate of the subject animals [82].
IL-10	IL-10 is an immunosuppressive cytokine consistently expressed in the tumor microenvironment. Studies carried out in different tumor models have demonstrated that blocking the IL-10R relieves immunosuppression in the tumor microenvironment and reinstates immune response directed at malignant cells [83].	This response was observed in an NB model wherein an antibody targeting the IL-10 receptor was used in combination with liposomal oligonucleotides to enhance the immune response. The observed immune response was larger compared with the use of oligonucleotides alone [84].
IFN-β	Interferon-beta reduces myeloid dendritic concentrations in peripheral blood. It also alters the function of dendritic cells and other APCs to downregulate antigen presentation and the ability of APCs to stimulate T cell responses [85].	IFN-β was found to increase the sensitivity of NB cells to the cytotoxic effects of the chemotherapy drug temozolomide (TMZ) through the mitigation of DNA repair enzyme (MGMT) expression [86].

**Table 4 ijms-24-08470-t004:** Ongoing clinical trials using combinational therapies of drugs with immunotherapeutic agents in the last ten years.

Identifier	Study Title	Phase	Start Date	Patients Enrolled	Status	Primary Aims
NCT02311621	Engineered Neuroblastoma Cellular Immunotherapy (ENCIT)-01	1	2014	65	Active, not recruiting	This is a phase 1 study designed to determine the maximum tolerated dose of the CAR+ T cells designed to recognize CD171 in patients with neuroblastoma.
NCT02914405	Phase I Study of 131-I mIBG Followed by Nivolumab and Dinutuximab Beta Antibodies in Children with Relapsed/Refractory Neuroblastoma (MiniVan)	1	2016	36	Recruiting	To determine the safety and tolerability of the novel combination of 131-I-MIBG, ch14.18/CHO and nivolumab in pediatric patients. This is assessed by the nature, frequency, severity, and timing of adverse events, including serious adverse events and immune-related adverse events, during the administration of ch14.18/CHO.
NCT033633	Naxitamab for High-Risk Neuroblastoma Patients with Primary Refractory Disease or Incomplete Response to Salvage Treatment in Bone and/or Bone Marrow	2	2017	122	Recruiting	The purpose of this study is to test the safety and efficacy of the combined therapy of naxitamab, a humanized monoclonal antibody targeting GD2 and GM-CSF in high-risk neuroblastoma patients.
NCT03294954	GD2 Specific CAR and Interleukin-15 Expressing Autologous NKT Cells to Treat Children with Neuroblastoma (GINAKIT2)	1	2017	36	Active, not recruiting	The purpose of this study is to find the largest effective and safe dose of GD2-CAR NKT cells (GINAKIT cells), to evaluate their effect on the tumor, how long they can be detected in the patient’s blood and what affect they have on the patient’s neuroblastoma.
NCT04049864	DNA Vaccination Against Neuroblastoma	1	2019	12	Recruiting	This is pilot open-label study to evaluate the safety and immunogenicity of a DNA vaccine strategy in relapsed neuroblastoma patients following chemotherapy and HSC transplantation.
NCT02395666	Preventative Trial of Difluoromethylornithine (DFMO) in High-Risk Patients with Neuroblastoma That is in Remission	2	2020	140	Active, not recruiting	To evaluate the preventative activity of DFMO as a single agent in patients that are in remission based on event-free survival (EFS).
NCT04637503	4SCAR-T Therapy Targeting GD2, PSMA and CD276 for Treating Neuroblastoma	1 and 2	2020	100	Recruiting	The purpose of this clinical study is to assess the feasibility, safety and efficacy of the combinational GD2, PSMA and CD276 4SCAR-T cell therapy against NB.
NCT05027386	Apatinib Mesylate Combined with IT Regimen for the Treatment of Recurrent or Refractory Pediatric Neuroblastoma	2	2021	62	Recruiting	The enrolled patients diagnosed with recurrent or refractory pediatric neuroblastoma received apatinib combined with IT regimen chemotherapy, the treatment includes a combination therapy phase and monotherapy maintenance phase.

## Data Availability

Not applicable.

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
