# Peer review of "Biological Insight and Recent Advancement in the Treatment of Neuroblastoma"

_ijms, 2023, doi:10.3390/ijms24108470_

Round 1

Reviewer 1 Report

This review article discusses about the recent advances in the treatment of neuroblastoma. My comments are as follows:

Lines # 59, 89 are not clear, hot tumors are “not” infiltrated by CD8+ T cells, cold tumors are not completely devoid of the immune effector cells rather infiltration level is low compared to hot tumors.

Line#91 Abbreviate TIL, it could be CD4+ T or NK cell also, therefore it is not reasonable to emphasize on CD8+ T cells

Cytokine section provide limited insights about their clinical relevance in neuroblastoma, authors also missed IL1b. I suggest expanding this section to include clinical trials conducted using these cytokines and their outcome.

Strategies discussed how neuroblastoma inhibits immune system are common to most of the solid tumors, there are plenty of review available. It would be a great addition to discuss about unique strategies utilized by neuroblastoma cells to evade immune system (PMID: 33341446).

Status of immune check point molecules on these cells and their role in immune escape and immune checkpoint inhibitors in clinical trials.

TGF-B paragraph, how it promotes cancer progression? Is it through Tregs??

Line#178, GM-CSF can be immunosuppressive (PMID: 35865534), it is one of the major factors associated with cytokine release syndrome in CAR-T cell therapy. I recommend the authors to cite this important review article about GM-CSF.

Line# 159, “there is also a larger quantity of Tregs during this earlier time of life, REF 35, please cross check this reference for the correctness of the statement.

Author Response

This review article discusses about the recent advances in the treatment of neuroblastoma. My comments are as follows:

Lines # 59, 89 are not clear, hot tumors are “not” infiltrated by CD8+ T cells, cold tumors are not completely devoid of the immune effector cells rather infiltration level is low compared to hot tumors.

Ans – We appreciate the reviewer bringing this to our attention; the statement has been corrected in the version.

Line#91 Abbreviate TIL, it could be CD4+ T or NK cell also, therefore it is not reasonable to emphasize on CD8+ T cells/

Ans – We appreciate the reviewer for pointing this out and letting us know. We have added acronyms to the updated MS.

Cytokine section provide limited insights about their clinical relevance in neuroblastoma, authors also missed IL1b. I suggest expanding this section to include clinical trials conducted using these cytokines and their outcome.

Ans – Since there have been so much research on the effectiveness of different cytokines in neuroblastoma, our evaluation did not specifically focus on cytokines' involvement. The involvement of numerous cytokines in the pathogenesis of neuroblastoma, however, have now been introduced as a new part to our website.

Strategies discussed how neuroblastoma inhibits immune system are common to most of the solid tumors, there are plenty of review available. It would be a great addition to discuss about unique strategies utilized by neuroblastoma cells to evade immune system (PMID: 33341446).

Ans - We totally agree with the critique, but we paid particular attention to the efficacy and comprehension of neuroblastoma treatments in our study. The immune system is also the subject of numerous evaluations, as the reviewer pointed out. Right now, it seems to fall outside the purview of this assessment. We added a new section on immunotherapy in neuroblastoma to the updated MS in order to give our review with novel viewpoints in the field.

Status of immune check point molecules on these cells and their role in immune escape and immune checkpoint inhibitors in clinical trials.

Ans – We thank the reviewer for their informative comments, and as the reviewer may have noticed, we have added a table (table 2) showing the effectiveness of immune checkpoint inhibitors.

TGF-B paragraph, how it promotes cancer progression? Is it through Tregs??

Ans - Thank you for your comment; however, we already covered how the Tregs affect NK function in the same line.

Line#178, GM-CSF can be immunosuppressive (PMID: 35865534), it is one of the major factors associated with cytokine release syndrome in CAR-T cell therapy. I recommend the authors to cite this important review article about GM-CSF.

Ans – We appreciate the reviewer's input, and we've updated our review to include the suggested reference.

Line# 159, “there is also a larger quantity of Tregs during this earlier time of life, REF 35, please cross check this reference for the correctness of the statement.

Ans – Though the stated reference was accurate, we nonetheless inserted a new reference for the readers' convenience and are grateful to the reviewer for checking this for us.

Reviewer 2 Report

Reviewer comments

Carlos Escutia and the group have discussed the “Biological Insight and Recent Advances in Treatment of Neuroblastoma” in the present article.  The content in the main text body is different from the abstract. The abstract should be modified based on the text of the manuscript. The structure and content must be revised, and results have to be better explained by the authors. Hence thorough revision should be needed to improve the manuscript based on the below comments. Overall, the paper is interesting, but the structure is confused. The theoretical framework is weak, and some results create confusion. Structure of the paper has to be improved; the study design, discussion, and presentation of results have to be clarified before being reconsidered for publication using suggested comments.

A major revision is required.

Major comment

1.      The title is “Biological Insight and Recent Advances in Treatment of Neuroblastoma” however there is no discussion about any treatment in the main text body in MS.

2.      In the abstract “The MYCN gene and GD2 surface antigen are two of the most useful immunotherapy findings for neuroblastoma” is mentioned however in the whole main text there is no discussion about MYCN & GD2 as a treatment for neuroblastoma.

3.      “This review attempts to identify important immunological actors and biological pathways involved in the dynamic interaction between the TME and immune system”, however, there is no biological pathway is discussed in MS.

4.      At least add 2-3 figures related to neuroblastoma disease and a figure showing the mechanism of treatment.

5.      The manuscript is mainly descriptive. Not many details about the mechanisms and treatments are described.

6.      Add more specific content with proper citations to enhance the quality of the paper.

7.      Table 2 title is not mentioned in MS and not cited in the main text body.

8.      Citation is written in the wrong format. Citations should be provided exactly in the journal’s specific format. Please check the citation style on the IJMS journal website.

9.      All references are written in the wrong format. References should be provided strictly in the journal’s specific design. Please check the reference style on the IJMS journal website.

Minor comments

1.      Acronyms of MYCN gene, GD2, NK, and NB cells.

2.      Remove colon (:) from line no 79.

3.      Table 1 title is not suitable.

4.      Table 1 and 2 is not cited in the main text.

5.      Table 2 lakes the citation.

6.      Line 128, check citation 24, I think it’s a copy-paste mistake.

7.      Line 137, T regulatory (Treg) Should be written in the same format. Check lines no 137, 159, and others. Acronyms should be written for the very first time in the text (line 135).

8.      In line 185, what is BMT?

9.      Reference: Only 1 citation from the year 2023 and 4 citations from the year 2022 however hundreds of papers are published in neuroblastoma in the last 2-3 years.

Author Response

  1. The title is “Biological Insight and Recent Advances in Treatment of Neuroblastoma” however there is no discussion about any treatment in the main text body in MS.

Ans – We appreciate the reviewer's input, and the updated version of the MS now reflects significant modifications. Table 1 also highlights the neuroblastoma treatment strategy.

  1. In the abstract “The MYCN gene and GD2 surface antigen are two of the most useful immunotherapy findings for neuroblastoma” is mentioned however in the whole main text there is no discussion about MYCN & GD2 as a treatment for neuroblastoma.

Ans – We appreciate the reviewer's feedback, and we did mention the function of the GD2 surface antigen in our initial MS. However, to emphasize the Anti-GD2 therapy, we have since included a few more parts. In the updated MS, we have also incorporated a new section on MYCN gene mutation.

  1. “This review attempts to identify important immunological actors and biological pathways involved in the dynamic interaction between the TME and immune system”, however, there is no biological pathway is discussed in MS.

Ans – Our goal was to present an overview of the TME and immunological agents. We have, however, made several significant adjustments in the updated MS because we concur with the reviewer.

  1. At least add 2-3 figures related to neuroblastoma disease and a figure showing the mechanism of treatment.

Ans – Thank you for the recommendation; to support the MS, we have added two figures and a new table.

  1. The manuscript is mainly descriptive. Not many details about the mechanisms and treatments are described.

Ans – Although it was not our intention to describe a specific method for this mechanism, we have added a brand-new section to the review to give it some structure.

  1. Add more specific content with proper citations to enhance the quality of the paper.

Ans - We appreciate the reviewer's concern and feedback, and we have updated the MS with more thorough information.

  1. Table 2 title is not mentioned in MS and not cited in the main text body.

Ans - The title for Table 2 has been added to the updated MS.

  1. Citation is written in the wrong format. Citations should be provided exactly in the journal’s specific format. Please check the citation style on the IJMS journal website.

Ans – The format of the MS has been altered. Formatting is typically done when a paper is accepted. Nowadays, sending a pre-formatted version for reference is pretty common.

  1. All references are written in the wrong format. References should be provided strictly in the journal’s specific design. Please check the reference style on the IJMS journal website.

Ans – Yes, when it was sent out for review, the MS was not formatted. To the best of my knowledge, the MS is formatted in accordance with the journal's guidelines when it is accepted.

Minor comments

  1. Acronyms of MYCN gene, GD2, NK, and NB cells.
  2. Remove colon (:) from line no 79.
  3. Table 1 title is not suitable.
  4. Table 1 and 2 is not cited in the main text.
  5. Table 2 lakes the citation.
  6. Line 128, check citation 24, I think it’s a copy-paste mistake.
  7. Line 137, T regulatory (Treg) Should be written in the same format. Check lines no 137, 159, and others. Acronyms should be written for the very first time in the text (line 135).
  8. In line 185, what is BMT?
  9. Reference: Only 1 citation from the year 2023 and 4 citations from the year 2022 however hundreds of papers are published in neuroblastoma in the last 2-3 years.

Ans – We appreciate the reviewer pointing out these little corrections, and we have now fixed all of the issues and updated the MS.

Round 2

Reviewer 1 Report

I am a bit disappointed to see that the revised version of this manuscript is more disorganized. Authors have introduced unnecessary vague terminology and headings. There are many formatting errors like size of the text between tables and main text is different (#table1 and 3, line #63-64, 169-170,199-200..). Although the new figure 2 seems useful, the resolution is too poor to read any word in the illustration. I strongly recommend the authors to focus on the recent advancement in the treatment of neuroblastoma.

1.    Table 1 provides limited insights about the recent advancement in the treatment of neuroblastoma. As I can see this information is scattered under different headings and can be combined so that a reader does not get confused. In recent advancement in treatment modalities one can expect how over the period clinical outcome improved in pediatric and adult patients. A table showing different drugs/immunotherapies in clinical trials e.g. CAR-T cells against CD171, B7H3, targets, Antibody mediated therapy, GM-CSF based vaccine etc.  

2. “The pediatric immune system fails to stop neuroblastoma” This heading is misleading and can be changed to mechanisms of immunosuppression in neuroblastoma.

3.    Why are there two different tables for cytokines? When we discuss tumor immune microenvironment, we should discuss all the immune players and how they mediate or inhibit the immune surveillance, not only the cytokines (table 2 and 3).

4.    In table 3 under heading “Cytokines in Neuroblastoma Pathogenesis & Preclinical Therapy” CCL2, CXCL12 and CXC3CL1 are not cytokines, they are chemokines.

5.    Headings like “Adult vs. Pediatric Immunology” are confusing. Information provided in line # 307-329 is not relevant to this section.

6.    GM-CSF plays an important role in immune activation and immunosuppression, important citations on GM-CSF are missing (e.g. PMID: 35865534).

Author Response

I am a bit disappointed to see that the revised version of this manuscript is more disorganized. Authors have introduced unnecessary vague terminology and headings. There are many formatting errors like size of the text between tables and main text is different (#table1 and 3, line #63-64, 169-170,199-200..). Although the new figure 2 seems useful, the resolution is too poor to read any word in the illustration. I strongly recommend the authors to focus on the recent advancement in the treatment of neuroblastoma.

Ans : We are feeling awkward that the reviewer is disappointed. We want to assure the reviewer that we will take utmost care of formatting the MS and the MDPI team will make sure that the MS is properly formatted to be published on the webpage. Due to track change, sometimes it’s tough to see the formatting of the MS.

  1. Table 1 provides limited insights about the recent advancement in the treatment of neuroblastoma. As I can see this information is scattered under different headings and can be combined so that a reader does not get confused. In recent advancement in treatment modalities one can expect how over the period clinical outcome improved in pediatric and adult patients. A table showing different drugs/immunotherapies in clinical trials e.g. CAR-T cells against CD171, B7H3, targets, Antibody mediated therapy, GM-CSF based vaccine etc.  

Ans: We completely agree with the reviewer, we were contemplating to add the table for clinical trial, as there are many publications out there citing the new ongoing clinical trial. However, in the revised MS we have added table 4 for the ongoing clinical trials with the combination of drugs and immunotherapy.

  1. “The pediatric immune system fails to stop neuroblastoma” This heading is misleading and can be changed to mechanisms of immunosuppression in neuroblastoma.

Ans: The heading has been changed to “Immunosuppressive pathways in Neuroblastoma” 

  1. Why are there two different tables for cytokines? When we discuss tumor immune microenvironment, we should discuss all the immune players and how they mediate or inhibit the immune surveillance, not only the cytokines (table 2 and 3).

Ans: It's a good question, however both tables highlight two separate components of the tumor microenvironment; in table 2, we discuss the TME in relation to ICI expression. Table 3 lists the cytokines and chemokines in neuroblastoma pathogenesis and preclinical therapy, and to the best of our knowledge, no review paper has shown the interaction with ICI.

  1. In table 3 under heading “Cytokines in Neuroblastoma Pathogenesis & Preclinical Therapy” CCL2, CXCL12 and CXC3CL1 are not cytokines, they are chemokines.

Ans : We are thankful to the reviewer for pointing out this error, we have fixed and have changed the title or table 3

  1. Headings like “Adult vs. Pediatric Immunology” are confusing. Information provided in line # 307-329 is not relevant to this section.

Ans : We have change the heading of the section. 

  1. GM-CSF plays an important role in immune activation and immunosuppression, important citations on GM-CSF are missing (e.g. PMID: 35865534).

Ans: We are thankful to the reviewer for suggesting very important reference which is now added in the MS.

Reviewer 2 Report

1. References should be cross-checked once specially in journal style.

2. No more comments.

Author Response

  1. References should be cross-checked once specially in journal style.

Ans: - We have double checked all the references, its tough to incorporate all the references in one review article.

  1. No more comments.

Ans: - Thank you.

Round 3

Reviewer 1 Report

I have no additional comments.